# Reducing the effective dosage of flutamide on prostate cancer cell lines through combination with selenium nanoparticles: An in-vitro study

**Iman Menbari Oskouie[1], Fatemeh Khatami[1], Amin Shiralizadeh Dezfuli[2],
Rahil Mashhadi[1], Akram Mirzaei[1], Seyedeh Negin Hashemi Dougaheh[1],
Helia Azodian Ghajar[1], Ramin Heshmat[3]\*, Seyed Mohammad Kazem Aghamir[1]\***

**1** Urology Research Center, Tehran University of Medical Sciences, Tehran, Iran, **2** Ronash Technology Pars Company (AMINBIC), Tehran University Science and Technology Park, North Campus of Tehran University, Tehran, Iran, **3** Chronic Diseases Research Center, Endocrinology and Metabolism Population Sciences Institute, Tehran University of Medical Sciences, Tehran, Iran

\* mkaghamir@tums.ac.ir (SMKA); rheshmat@tums.ac.ir (RH)

## Abstract

### Objective

Objective of the study was to evaluate the therapeutic potential of selenium nanoparticles (SeNPs) in combination with flutamide for treating prostate cancer (PCa) cell lines. The goal was to reduce the dosage of flutamide to decrease its side effects, especially hepatotoxicity.

### Materials and methods

PC3, LnCAP, and DU145 cell lines were treated with varying concentrations of SeNPs and Flutamide to determine IC50 values using the MTT assay. Subsequently, the IC50 concentration of flutamide was reduced by 50% and different concentrations of SeNPs were added to determine new IC50 concentrations of the combinations. Annexin-V/ PI staining was performed to assess the apoptosis rate. The DNA cell cycle was analyzed using the PI staining technique. Migration, proliferative capability, and nucleus morphology of the cells were evaluated through the scratch-wound assay, colony-forming assay, and Hoechst staining, respectively. The expression of *SNAIL*, *KLK3*, *E-cadherin*, *VEGF-C*, *HIF-1α*, *Bcl2*, and *BAX* were examined using real-time PCR.

### Results

All treated groups significantly increased early and late apoptosis rate of the PCa cell lines, and induced SubG1/G1 arrest in the cell cycle assay, compared to the control group. Significant inhibition of migration potential and colony formation was observed in all treated groups. Our results suggest that the combination group (50% decrease of Flutamide dosage) treatment upregulated apoptosis-related genes and *KLK3*, and downregulated genes involved in angiogenesis and proliferation similar to Flutamide alone (p > 0.05).

**Data availability statement:** All relevant data are within the paper and its Supporting Information files.

**Funding:** The author(s) received no specific funding for this work.

**Competing interests:** There is no competing interest.

## Conclusion

It is suggested that simultaneous administration of SeNPs and flutamide could potentially reduce the effective dosage of flutamide and decrease its adverse effects.

## 1. Introduction

Men between the ages of 45 and 60 are commonly affected by prostate cancer (PCa), which accounts for the majority of cancer-related deaths in Western countries [1]. According to GLOBOCAN 2020, there were 1,414,259 new cases of prostate cancer and 375,304 deaths attributed to this disease worldwide [2].

Various methods such as, prostate-specific antigen (PSA) testing, digital rectal examination (DRE), magnetic resonance imaging (MRI), or health screening, are used to detect PCa in many males [3]. Therapeutic modalities for PCa include active surveillance, chemotherapy, radiation therapy, hormone therapy, surgery, and cryotherapy [4].

Androgen-deprivation therapy (ADT), commonly known as hormonal treatment, is suggested for the treatment of recurrent malignancies that have metastasized outside the prostate [1]. Flutamide is a nonsteroidal medication that inhibits androgen receptors without hormonal agonist effects. The combination of flutamide and an LHRH agonist has yielded encouraging outcomes in treating PCa. In in-vitro experiments, flutamide has been demonstrated to act as an antagonist specifically in the ventral prostate and androgen-dependent seminal vesicles [5,6].

Each derivative of selenium has been documented to possess distinct mechanisms of action that exhibit anticancer properties, with a majority of them specifically noted for their ability to suppress PCa. For example, it has been observed that Selenite induces caspase-mediated death in PC-3 cells, accompanied by DNA breakage, activation of *JNK1/2* and p38 *MAPK/SAPK2*, as well as the formation of mitochondrial superoxide [7,8]. Additionally, it has been documented that it leads to the interruption of the G2/M cell cycle and triggers programmed cell death in HCT116 and SW620 colorectal cancer cells via the Bax-dependent mitochondrial space [8]. Nevertheless, selenium nanoparticles (SeNPs) have garnered significant interest due to their distinctive optical, magnetic, and structural characteristics that surpass those of molecules or bulk solids [9,10]. Also, SeNPs can serve as carriers for drugs or medicinal substances in cancer treatment. These nanomaterials have excellent biocompatibility, low toxicity, rapid absorption, and exhibit significant effects [11–15]. However, studies have shown that chemically produced nanosized selenium can induce cell cycle arrest, specifically at the S phase, in HeLa cells [16]. It also hinders the development of LNCaP cells by reducing the production of androgen receptors in both the gene transcription and protein synthesis sites. Furthermore, it activates the phosphorylation and androgen receptor degradation via the *Akt/Mdm2* route [17].

Hepatotoxicity, leukopenia, and the growth of medication tolerance are negative impacts that are linked to every PCa therapy [18]. While it has become widely recognized that flutamide causes hepatic impairment, an examination of the drug's combination with antiandrogen medication suggested that flutamide would be effective if frequent liver functionality tests were conducted while the patient was receiving treatment [19]. Therefore, it is crucial to discover new, affordable chemotherapeutic drugs with little to no side effects and improved effectiveness.

Combined therapy is particularly intriguing for cancer medical care, mainly in males with progressed PCa. We hypothesized that the combination of flutamide with SeNPs would augment the anticancer potency by concurrently targeting multiple molecular sites. Additionally, it has the potential to result in reduced therapeutic dosages of Flutamide and decrease the

hepatotoxicity side effects associated with Flutamide. Hence, we aimed to assess the combined impact of SeNPs and Flutamide on the growth and programmed cell death, as well as the expression of various genes such as *SNAIL*, *KLK3*, *E-cadherin*, and apoptotic genes in three types of prostate cancer cells: LNCaP, PC3, and DU145.

## 2. Materials and methods

The study was conducted at the Urology Research Center, Tehran University of Medical Science. Subsequent to obtaining REB authorization from the Ethics Review Board of Sina Hospital affiliated with Tehran University of Medical Sciences. (IR.TUMS.MEDICINE. REC.1401.499).

### 2.1. Preparation and characterization of SeNPs

The compounds, including sodium selenite ($Na_2SeO_3$, CAS number: 10102-18-8), polyvinylpyrrolidone (PVP, CAS number: 9003-39-8), and sodium hydroxide (NaOH, CAS number: 1310-73-2), were acquired from Sigma Aldrich USA. These substances were utilized as obtained, without any further modification. Distilled water was used in this experimental study.

The SeNPs were produced using the subsequent method. 1 g of PVP was blended with a 60 ml solution of distilled water. This mixture was then mixed with 40 ml of a solution containing 100 mM concentration of $Na_2SeO_3$. The pH of the mixture was brought to 7.1 by adding 1.0 M NaOH. The resulting red solution underwent dialysis in double-distilled water for 72 hours, with water changes every 24 hours. The SeNPs-containing solution was kept at a temperature of 4°C for further investigations.

Transmission electron microscopy (TEM; Zeiss-EM10C) was employed to analyze the size and shape of the SeNPs at an 80 kV accelerating voltage. To obtain absorption spectra in the 200–1000 nm wavelength range, an ultraviolet-visible (UV-vis) spectrophotometer with a 2 nm resolution was used. The SeNPs were also studied through Dynamic Light Scattering (DLS, Particle sizing systems, USA) using a NICOMP 380/ZLS (PSS) system, with the data processed by ZPW388 software.

### 2.2. Cell lines, and cell culture

The LNCaP (ATCC Number: CRL-10995, NCBI Code: C439), PC3 (ATCC Number: CRL-1435, NCBI Code: C427), and DU145 (ATCC Number: HTB-81, NCBI Code: C428) prostate cancer cell lines were obtained from the National Cell Bank of Pasteur Institute in Tehran, Iran. These cells were cultured in DMEM medium (Gibco, Carlsbad, CA) supplemented with 10% heat-inactivated fetal bovine serum (FBS, Gibco, Carlsbad, CA), 100 units/ml Penicillin, 2 mM L-glutamine, and 100 μg/ml Streptomycin (Gibco BRL, Grand Island, NY). The culturing conditions included a 37°C temperature and a humidified atmosphere containing 5% $CO_2$. A control group was set up using DMEM media with 0.1% dimethyl sulfoxide (DMSO) (Sigma-Aldrich, St. Louis, MO, USA). For the treatment of these cell lines, Flutamide was dissolved in DMSO and sterile phosphate-buffered saline (PBS) to prepare a stock solution.

### 2.3. Analysis of cell survival

The MTT test was employed to quantify the suppressive impact of Flutamide and SeNPs on the metabolic activity of PC3, LNCaP, and DU145 cell lines. We conducted the MTT assay in triplicate across three independent experiments to ensure reproducibility and reliability of the results. For each drug concentration tested, triplicate wells were used in each experiment. This allowed us to accurately assess the IC50 value by averaging the triplicate values and analyzing

the data across the three separate experimental runs. All cell lines were seeded at a density of $5 \times 10^3$ cells per well in 96-well plates and incubated for 24, 48, and 72 hours. Plates were treated with different concentrations of Flutamide (ranging from 3 to 30 μM, with each dosage increasing by 25%) and SeNPs (ranging from 300 to 3000 μM, with each dosage increasing by 25%), both individually. Incubation was carried out at 37 °C with 5% $CO_2$ saturation. To prepare the MTT solution, 5 milligrams of 3-[4,5-dimethylthiazol-2-yl]-2,5 diphenyl tetrazolium bromide powder (Sigma-Aldrich, St. Louis, MO, USA) fully dissolved in 1 milliliter of sterile PBS to obtain a concentration of 5 milligrams per milliliter. Next, dilute with the sterile PBS to a final volume of 10 ml, resulting in a sample solution with a concentration of 0.5 mg/ml. This solution should be freshly prepared for each usage. Subsequently, during a 24, 48, and 72-hour period, 100 μL of the MTT solution (0.5 mg/mL) is introduced to the cells. The cells will then be subjected to a 4-hour incubation at 37 °C to facilitate the metabolization of this solution. Following the disintegration of the formazan crystals in 100 ml of DMSO, a mixture with a purple color will result. An ELISA microplate reader (MPR4+, Hyperion, Medizintechnik GmbH & Co.KG, Germany) was used for measuring the optical density at a range of 570 nm.

For ascertaining the half maximum inhibitory concentration (IC50) of flutamide and SeNPs for the PC3, LNCaP, and DU145 cell lines, the test was carried out triple times. Graphs depicting dose and time responses were created, and the IC50 value, indicative of the concentration that reduces the growth of the control cells by 50%, was calculated using GraphPad PRISM software version 9 (San Diego, CA). The formula for calculating the percentage of cells using the MTT technique is as follows:

$$\frac{\left(\text{Mean absorbance measured from three replicate drug wells}\right)}{\text{Mean absorbance obtained from the control wells}} \times 100\% = \% \; cell \; viability$$

After calculating the IC50 concentration of SeNPs and flutamide following 24, 48, and 72 hours of exposure to drugs in various PCa cell lines, the IC50 concentration of flutamide was decreased by 50%, and 75%. Different concentrations of SeNPs were then added to determine the IC50 concentration of flutamide when combined with SeNPs at different times and in different cell lines. This approach, allowed for the identification of these combinations that had a similar inhibitory effect compared to flutamide alone [20].

## 2.4. Assessing cell morphology and structure using crystal violet dye

$5 \times 10^4$ prostate cancer cells (LNCaP, PC3, and DU145) were seeded into six-well plates following a forty-eight-hour treatment with IC50 concentrations of Flutamide, SeNPs, and their combinations. The cell lines were rinsed two times with PBS and subsequently treated with Paraformaldehyde for fixation. Afterwards, the cells were stained using a 0.5% w/v solution containing violet crystal. Cell morphology was assessed by employing an inverted microscope.

## 2.5. In vitro 2D colony formation assay

To determine whether prostate cancer cells in a cell medium used for cultivation were invasive, a colony forming test was conducted. In a six-well growing medium, LNCaP, PC3, and DU145 prostate cancer cell lines were grown at a density of 1000 cells per well. Following a 48-hour incubation period, the cells were exposed to IC50 doses of Flutamide, SeNPs, and their combinations. The plate was then placed in an incubator at a temperature of 37°C until cells develop into colonies that can be seen. After a period of 14 days, the culture medium was rinsed with PBS solution two times. Subsequently, the colony cells were exposed to a 0.5% w/v crystal violet solution for a duration of 30 minutes at a temperature of 25°C. A colony was defined as consisting of 50 or more cells, while a cluster consisted of 3–50 cells. The

experiments were repeated three times. Photographs were taken of each well, and the quantity of colonies in every image were counted and assessed utilizing ImageJ software.

## 2.6. Staining of cells with Hoechst dye (33342)

An assessment of apoptosis in cancer cells was conducted utilizing a Hoechst dye test. Prostate cancer cells were seeded into a 96-well plate at a density of $2.5 \times 10^3$ cells per well. They were then treated with flutamide, SeNPs, and a combination of both. After a 48-hour incubation period, the cells were fixed with 100 μL of cold paraformaldehyde (4%) and subsequently rinsed two times with 100 μL of PBS. Next, a volume of 2 μL of Hoechst dye was introduced to the cell pellet and incubated at a temperature of 25 °C, while maintaining darkness. After 30 minutes, the sample was examined using a fluorescence microscope at 100X magnification to detect nuclear condensation and count the number of broken nuclei in intact cells. Apoptosis was identified by the presence of fragmented nuclei in certain cells.

## 2.7. Measurement of cell migration

In a six-well culture plate, LNCaP, PC3, and DU145 prostate cancer cell lines were grown at a density of $5 \times 10^5$ cells per well. Once the cells attained a density of 85%, a sampler tip was used to construct a vertical path across the diameter of the six cells. In order to eliminate the individual cells, the lower part of the incubation plate was delicately rinsed with PBS. After-ward, certain cells were subjected to PBS in order to serve as a control group, while others were treated with drugs. All cell lines received IC50 doses of flutamide, SeNPs, and a combi-nation of both. Subsequently, imaging was conducted at 0 and 48 hour intervals. The images were analyzed using ImageJ software, and wound closure was calculated using the following formula:

$$1 - \frac{\text{area of wound at day } 2}{\text{area of wound at day } 0} \times 100\% = \% \ wound \ closure$$

## 2.8. tumor spheroid model

Plates covered with agar were utilized to produce tumor spheroids. According to Friedrich et al. [21], 50 μl/well of agar (2% (wt/vol) dissolved in RPMI) was applied to 96 well plate. PC3, LnCap, and DU145 cells were seeded in wells with a seeding density of 2000 cells per well and left to cultivate at a temperature of 37 °C in an environment with 5% carbon dioxide. Follow-ing a duration of a 96 hour, a solitary spheroidal object with a diameter ranging from 300μm to 400μm had formed within every individual well. The spheroids from all cell lines were then treated with IC50 doses of flutamide, SeNPs, and their combinations. After 48 hours, the colonies were stained with 0.5% w/v crystal violet solution for 30 minutes at 25 °C, followed by two washes with PBS.

## 2.9. Evaluating cell apoptosis through flow cytometry

The cell apoptosis assessment was conducted utilizing the manufacturer's protocol for the fluorescein-conjugated annexin V (annexin V-FITC) staining assay. The LNCaP, PC3, and DU145 cell lines were placed in six-well dishes with a population density of $3 \times 10^5$ cells per well. Cells were subsequently cultured for 48 hours with and without IC50 doses of SeNPs, Flutamide, and their mixtures. After two rinses with PBS, 100 μl of staining solution compris-ing PI and annexin-V was added to the medicated and controlled prostate cancer cell popu-lations. The cells were then incubated for 15 minutes at 25°C in dark conditions. Afterward,

flow cytometry was used to evaluate the fluorescent signals. The scatter maps were divided into four sections. The area representing viable cells (annexin-V-negative and PI-negative) was labeled as Q4 and displayed in the lower left corner. The area representing early apoptotic cells (annexin-V-positive and PI-negative) was labeled as Q3 and demonstrated in the lower right quadrant. The region representing late apoptotic cells (annexin-V-positive and PI-positive) was labeled as Q2 and presented in the upper right quadrant. The zone representing necrotic cells (annexin-V-negative and PI-positive) was labeled as Q1 and exhibited in the upper left quadrant. The apoptotic rate was quantified as the proportion of annexin V+/PI- cells using a BD flow cytometer and analyzed with Flowjo software (Tree Star Inc., version 9.6.3, USA) [22,23].

## 2.10. DNA cell cycle analysis

Propidium Iodide (PI) staining was used to analyze the cell cycle. PC3, LNCaP, and DU145 cell lines were seeded in 6-well plates at a population density of $5 \times 10^5$ cells per well. The cells were then treated with flutamide, SeNPs, and their combination at IC50 doses for a duration of 48 hours. After treatment, the cells were rinsed two times with PBS, fixed with 70% cold ethanol, and stored at −20°C overnight. Subsequently, the cells were rinsed with PBS two times and subjected to a 30-minute incubation at 37°C with RNase I (100 μg/ml) and stained DNA with 500 μL PI (50 μg/ml in 0.1% Triton X-100/0.1% sodium citrate). A BD flow cytometer was used to separate the cells. A sample of DNA was assessed by flow cytometry, and the findings were evaluated using the Flowjo program (Tree Star Inc., version 9.6.3, USA). The hypodiploid sub-G0/G1 DNA percentage can be used to determine the apoptotic cell fraction.

## 2.11. RNA extraction, cDNA synthesis, and gene expression analysis by real-time PCR

In adherence to the guidelines provided by the manufacturer, total RNA was isolated utilizing the Highly Pure RNA isolation kit (Roche Applied Science, Germany). Initially, the quantification and purity of total RNA were evaluated by means of spectrophotometry at 260 and 280 nanometers with a Nanodrop ND-1000 (Nanodrop Technologies, Wilmington, DE). Following this, complementary DNAs (cDNAs) were reverse transcribed from 1 to 2 μg of total RNA using the cDNA synthesis PrimeScript RT reagent Kit (Takara Bio Inc., Otsu, Japan). According to the manufacturer's guideline, the cDNA level was adjusted for normalization through a series of PCR employing *B2M* primers (Table 1). Subsequently, the normalized cDNAs underwent production utilizing the real-time PCR cycler from QIAGEN. The housekeeping gene *B2M* was employed to establish normalization of gene expression levels, while the relative expression was computed using the $2^{-\Delta\Delta CT}$ approach. Table 1 presents the primers and their corresponding amplicon lengths.

## 2.12. Statistical analysis

The means ± SE of the triplicate determinants were used to display all the data. The data were assessed using ANOVA and t-tests. When compared to the matching control, *P < 0.05, **P < 0.01, ***P < 0.001, and ****P < 0.0001 were considered statistically significant.

## 3. Results

### 3.1. Characterization of SeNPs

The morphology of the SeNPs was examined through TEM. The SeNPs were spherical in shape and had various particle diameters, as displayed in Fig 1A. According to the TEM data,

**Table 1. Nucleotide sequences of primers used for real-time PCR.**

| Gene | Forward primer (5′ -3′) | Reverse primer (5′ -3′) |
| --- | --- | --- |
| KLK3 | CGTGACGTGGATTGGTGCT | TTCCTGATGCAGTGGGCAGC |
| E-cadherin | TCGTAACGACGTTGCACCAA | TTCGGAACCGCTTCCTTCAT |
| SNAIL | TAGCGAGTGGTTCTTCTGCG | AGGGCTGCTGGAAGGTAAAC |
| VEGF-C | GCTTCTTCTCTGTGGCGTGT | CTTTGCTTGCATAAGCCGTGG |
| Bcl2 | CCCCGCGACTCCTGATTCAT | CAGTCTACTTCCTCTGTGATGTTGT |
| BAX | CGGGTTGTCGCCCTTTTCTAC | AGTCCAATGTCCAGCCCATGA |
| HIF-1α | GTGCCACATCATCACCATATAG | GCTTTCTCTGAGCATTCTGCAA |
| B2M | TGTCTTTCAGCAAGGACTGGT | TGCTTACATGTCTCGATCCCAC |

the average diameter of the SeNPs is 45.34 nm. The distribution of selenium nanoparticles (depicted as dark spots) was monodispersed and exhibited a homogeneous structure.

Particle size measurements were determined using DLS analysis as shown in Fig 1B. The results revealed that the average values of particle size for SeNPs were 66.61 nm.

The dispersion of SeNPs was analyzed using UV-Vis spectrophotometry, revealing morphological alterations such as variations in nanoparticle dimensions and agglomeration (Fig 1C). It was determined that the development of surface plasmon vibration (SPR) on spherical SeNPs was responsible for the maximum absorption bands situated between 200 and 300 nm [24,25]. The diameter of the SeNPs is directly proportional to the wavelength at which the highest intensity of the SPR band occurs. Additionally, the total concentration of SeNPs is directly related to the content of the absorbance peak. A study found that glucose-stabilized Se nanospheres exhibited an absorption maximum at a wavelength of 295 nm, similar to our study [26].

## 3.2. effect of SeNPs and flutamide on cell proliferation

In PC3, LNCap, and DU145 cell lines, the cytotoxic effects of SeNPs and flutamide were investigated. Based on the results (Fig 2), the IC50 values after 48 hours of treatment for SeNPs were 1788 μM, 1430 μM, and 915 μM for PC3, LNCap, and DU145 cell lines, respectively. The IC50 values after 48 hours of treatment for Flutamide were 17.88 μM, 14.3 μM, and 11.44 μM for PC3, LNCap, and DU145 cell lines, respectively. The results exhibited that SeNPs and Flutamide had an important cytotoxic effect on all three cell lines in a dose-dependent way. To evaluate the synergistic characteristic of SeNPs and Flutamide the viability of treated cells was measured after treating prostate cancer cells with the combination of flutamide and SeNPs as described in the materials and methods section. The new IC50 values for combination therapy (50% decrease in flutamide IC50 concentration) after 48 hours of treatment were 223.5 μM SeNPs + 8.94 μM Flutamide for PC3, 178.8 μM SeNPs + 7.15 μM Flutamide for LNCap, and 114.4 μM SeNPs + 5.72 μM Flutamide for DU145. Additionally, IC50 values for combination therapy (75% decrease in flutamide IC50 concentration) after 48 hours of treatment were 1117 μM SeNPs + 4.47 μM Flutamide for PC3, 894 μM SeNPs + 3.57 μM Flutamide for LNCap, and 715 μM SeNPs + 2.86 μM Flutamide for DU145 (Fig 2). The IC50 concentrations for 48 hours were selected for all subsequent experiments. In the combination treatment group, the synergistic effect of SeNPs and Flutamide was more prominent when a 50% reduction in Flutamide IC50 concentration occurred. Therefore, these concentrations were chosen for combination treatment in the following experiments.

## 3.3. Morphological changes

Examining cellular morphology is crucial for comprehending cell attitude. After treating cells with flutamide, SeNPs, and their combinations for 48 hours, morphological alterations of

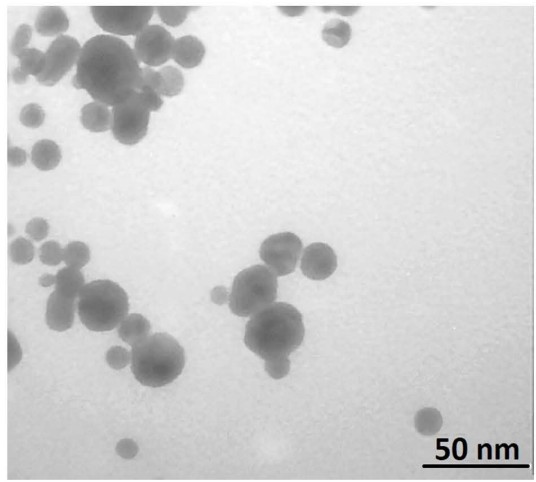

**A**

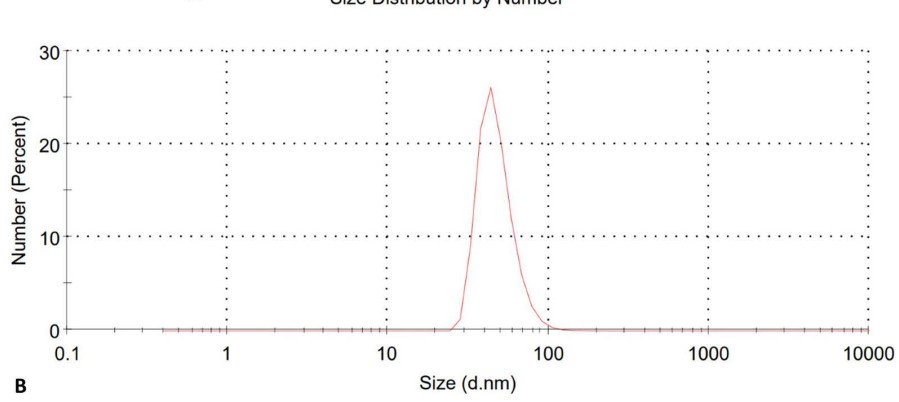

**B**

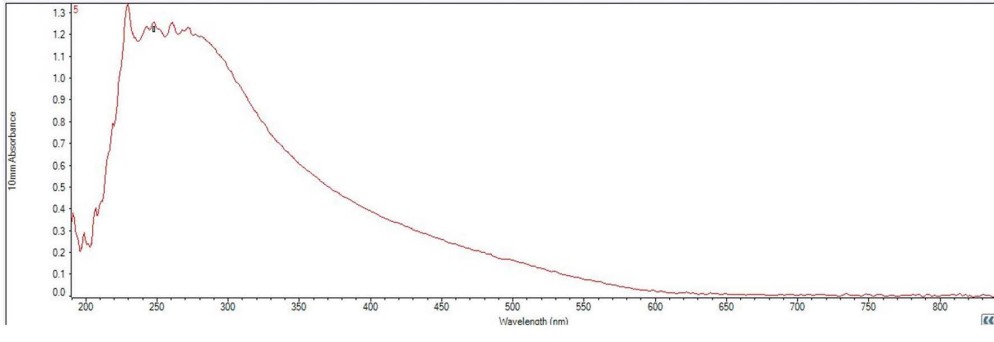

**C**

**Fig 1. Characterization of SeNPs.** (A): TEM image of SeNPs, (B): DLS of SeNPs, (C): UV-Vis spectrum of SeNPs.

PC3, LnCAP, and DU145 cell lines were examined using an inverted microscope. As depicted in Fig 3, the cells underwent morphological alterations such as shrinkage, rounding, elongation, membrane protrusion, and a decrease in the number of living cells after treatment with flutamide, SeNPs, and their combination in comparison to the control group. These structural changes imply that Flutamide and SeNPs have the potential to cause apoptosis in the PC3, LNCap, and DU145 cell lines.

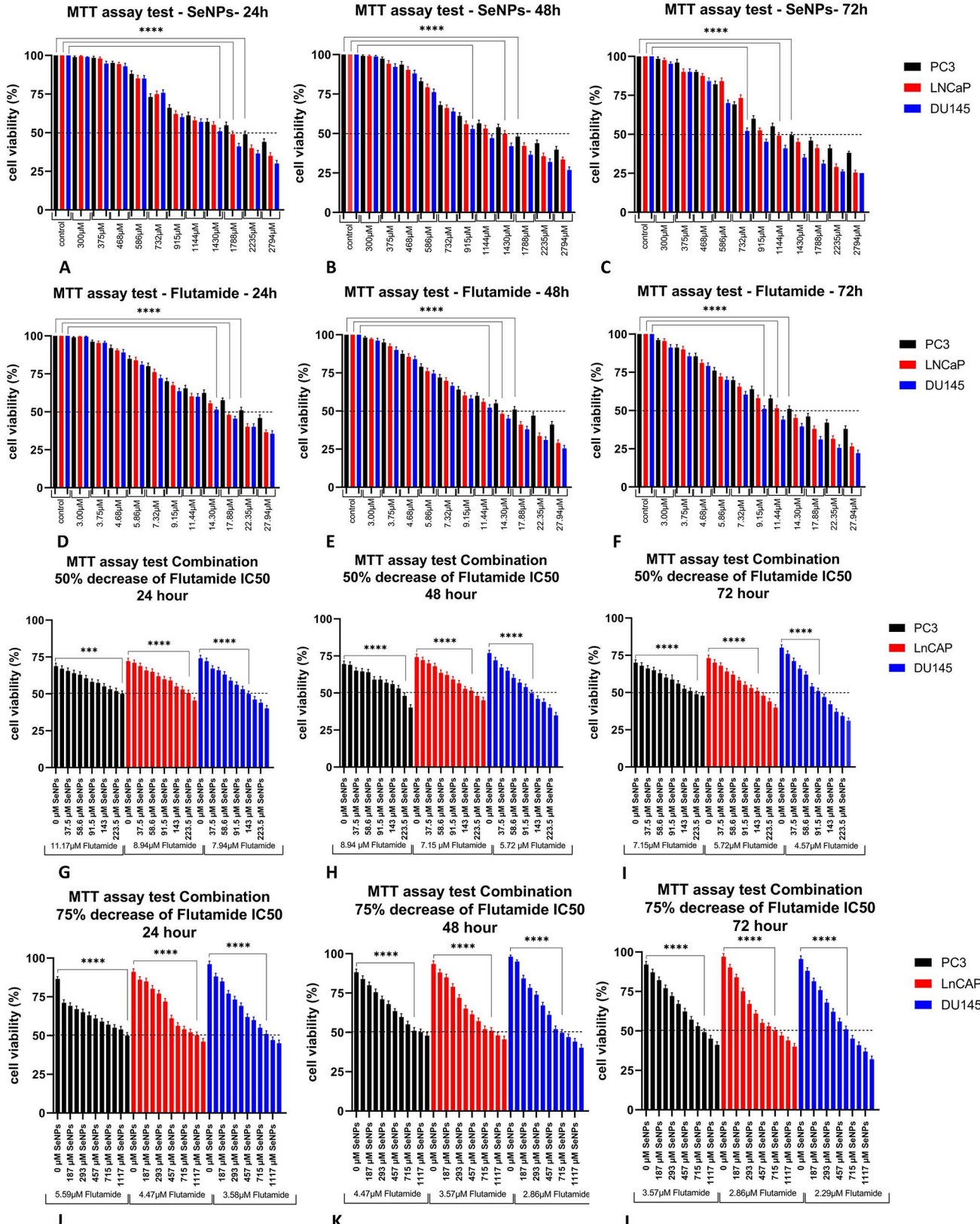

**Fig 2. The effect of SeNPs, flutamide, and their combinations with various concentrations on PC3, LNCap, and DU145 cell proliferation.** The anti-growth effect of SeNPs (A–C), Flutamide (D–F), and their combinations (G–L) after 1,2 and 3 days on PC3, LnCAP, and DU145 cell lines was assessed by MTT test.

### 3.4. Effects of SeNPs and flutamide on the nuclei of prostate cancer cell lines

The PC3, LNCaP, and DU145 cell lines were treated with IC50 doses of Flutamide, SeNPs, and their mixtures for a duration of 48 hours to evaluate apoptosis through fluorescent microscopy and Hochst staining techniques. Outcomes obtained using the Hoechst 33342 fluorescent dye are presented in Fig 4, where the nuclei in the control group showed blue fluorescence. After treatment with SeNPs and Flutamide, significant alterations in the nuclei morphology were seen. Several nuclei split and disintegrated following exposure to their IC50 concentrations of SeNPs, Flutamide, and their combination compared to the control group after 48 hours, resulting in dispersed nuclear content. Additionally, apoptosis characteristics were seen under a fluorescent microscope (Fig 4).

### 3.5. Effects of SeNPs and flutamide on the migration of the prostate cancer cells

Fig 5 and S1 and S2 Figs display the outcomes of the migration assay. As illustrated in Fig 5, the movement of prostate cancer cells was significantly reduced by treatment with SeNPs, Flutamide, and their mixture after 48 hours, in comparison to the control group. The percentage of wound closure in both the treated and control groups across all cell lines is depicted in Fig 5. Nevertheless, in the control group, the gap between cells has nearly closed.

### 3.6. Decrease in the number of colonies by Flutamide and SeNPs

There was a clear connection between the metastatic capability of the prostate cell lines and the establishment of colonies by untreated prostate cancer cells. The untreated PC3 cells exhibited a higher number of colonies compared to the LNCap and DU145 cell lines. Both SeNPs and Flutamide, when used at IC50 dosages, effectively suppressed the colony-establishing capability of all the cell lines, in comparison to the unattended cell lines. Additionally, a combination of the two compounds effectively halted the growth of colonies in all administered cell lines (Fig 6).

### 3.7. SeNPs and Flutamide destroy tumor spheroid

Three-dimensional *in vitro* simulations exhibit certain aspects of the structure of human tumors and demonstrate resistance to anti-cancer medications [27,28]. Therefore, we assessed the efficacy of SeNPs, Flutamide, and their combinations in a three-dimensional tumor spheroid model. PC3, LNCaP, and DU145 spheroids were cultured and allowed to grow until they reached a diameter of 300–400 µm. Spheroids that received IC50 doses of SeNPs, Flutamide, and their combinations began to shrink and release dead cells after 48 hours of exposure, while the untreated spheroids kept growing. The spheroids that underwent treatment exhibited substantial structural impairment and included a numerous deceased cells (Fig 7). In contrast, the spheroids that received no therapy maintained their normal and undamaged shape [29].

### 3.8. Induction of apoptosis by SeNPs and Flutamide

In order to ascertain the potential of SeNPs and Flutamide to induce apoptosis in prostate cancer cell lines, Annexin-V/PI staining was performed. Using flow cytometry, the apoptotic effects of SeNPs, Flutamide, and their combinations were analyzed. Without any treatment, the percentage of initial and final apoptotic cells in the PC3 cell line was 2.09% and 9.18%, respectively. For the LnCAP cell line, these values were 2.72% and 13.7%, and for the DU145

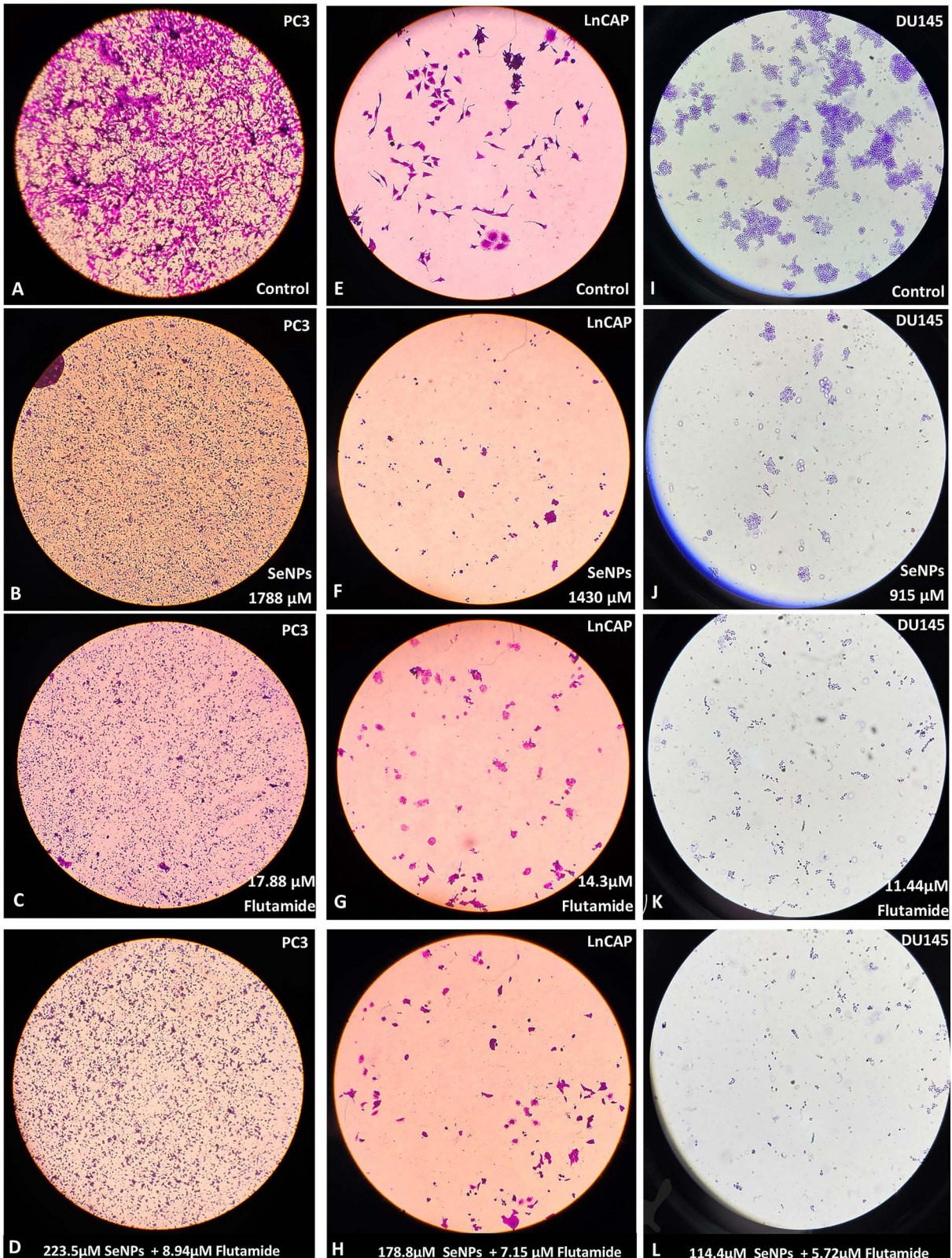

**Fig 3. Morphological changes in prostate cell lines after treatment with SeNPs, flutamide, and their combinations.** (A) PC3-control (B) PC3-SeNPs (C) PC3-Flutamide (D) PC3-combination (E) LnCap-control (F) LnCap-SeNPs (G) LnCap-Flutamide (H) LnCap-combination (I) Du145-control (J) Du145-SeNPs (K) Du145-Flutamide (L) Du145-combination.

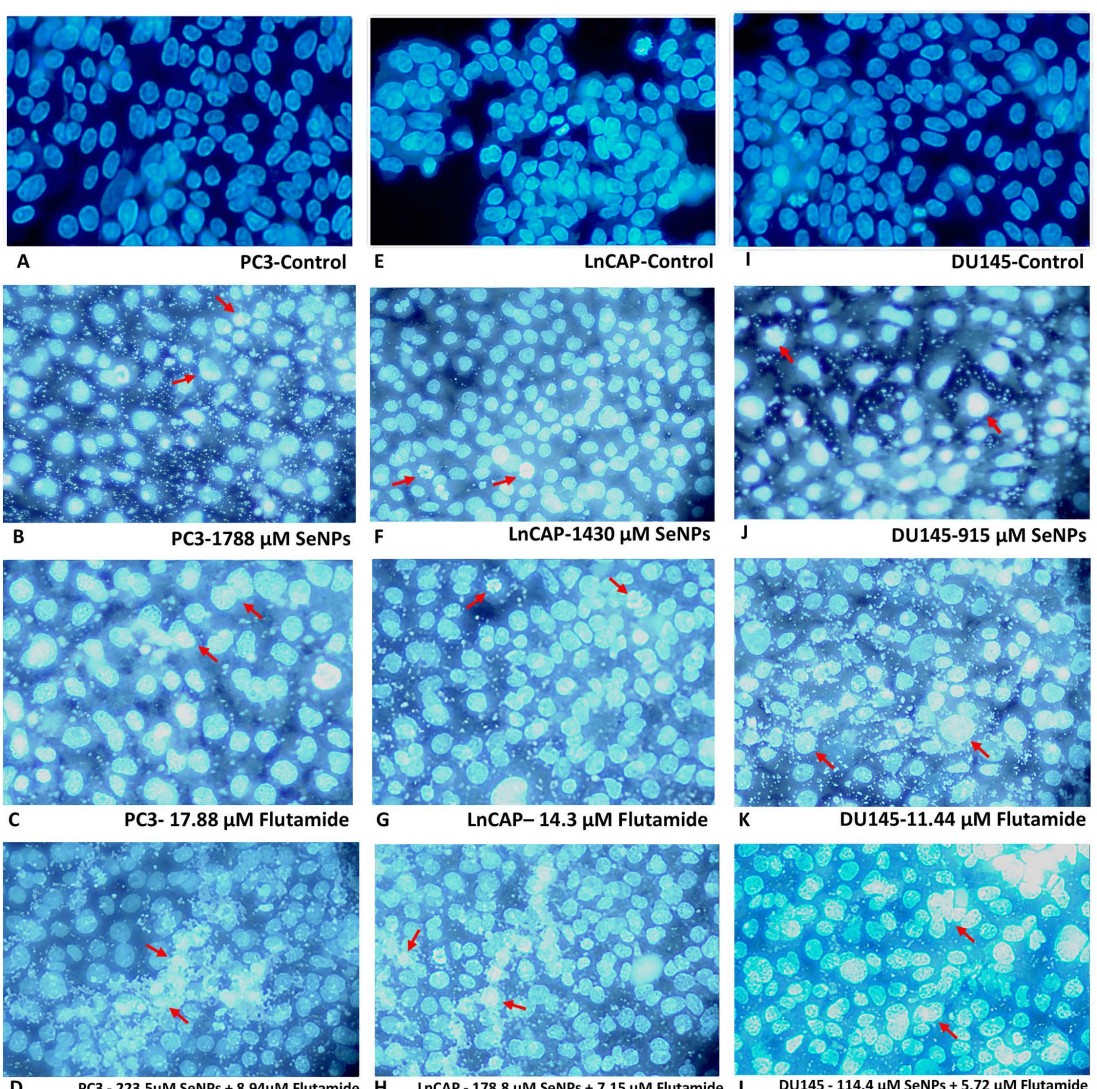

**Fig 4. Fluorescent staining with Hoechst dye (33342) of prostate cell line after treating with SeNPs, flutamide, and their combinations.** (A) PC3-control (B) PC3-SeNPs (C) PC3-Flutamide (D) PC3-combination (E) LnCap-control (F) LnCap-SeNPs (G) LnCap-Flutamide (H) LnCap-combination (I) Du145-control (J) Du145-SeNPs (K) Du145-Flutamide (L) Du145-combination. Fragmented nuclei in some cells indicate apoptosis.

cell line, they were 2.03% and 16.4%. However, when treated with SeNPs, these percentages increased to 3.37% and 51.8% for PC3 cells, 3.6% and 36.7% for LnCAP cells, and 2.34% and 45.7% for DU145 cells. The percentage of apoptotic cells after treatment with Flutamide was 6.59% and 33.5% for PC3 cells, 3.18% and 45.5% for LnCAP cells, and 5.28% and 49.3% for DU145 cells. The apoptosis percentages after treatment with the IC50 value of the combination of SeNPs and Flutamide were similar. Specifically, the percentage of apoptotic cells was 3.89% and 40.4% for PC3 cells, 2.49% and 34.7% for LnCAP cells, and 3.28% and 42.7% for DU145 cells. In the PC3 and LnCAP cell lines, the percentage of necrotic cells was greater in the group of cells that had been administered flutamide. Finally, the extremely small percentage of necrotic cells in SeNPs treated cells suggests that this medication is likely safe for use (Fig 8).

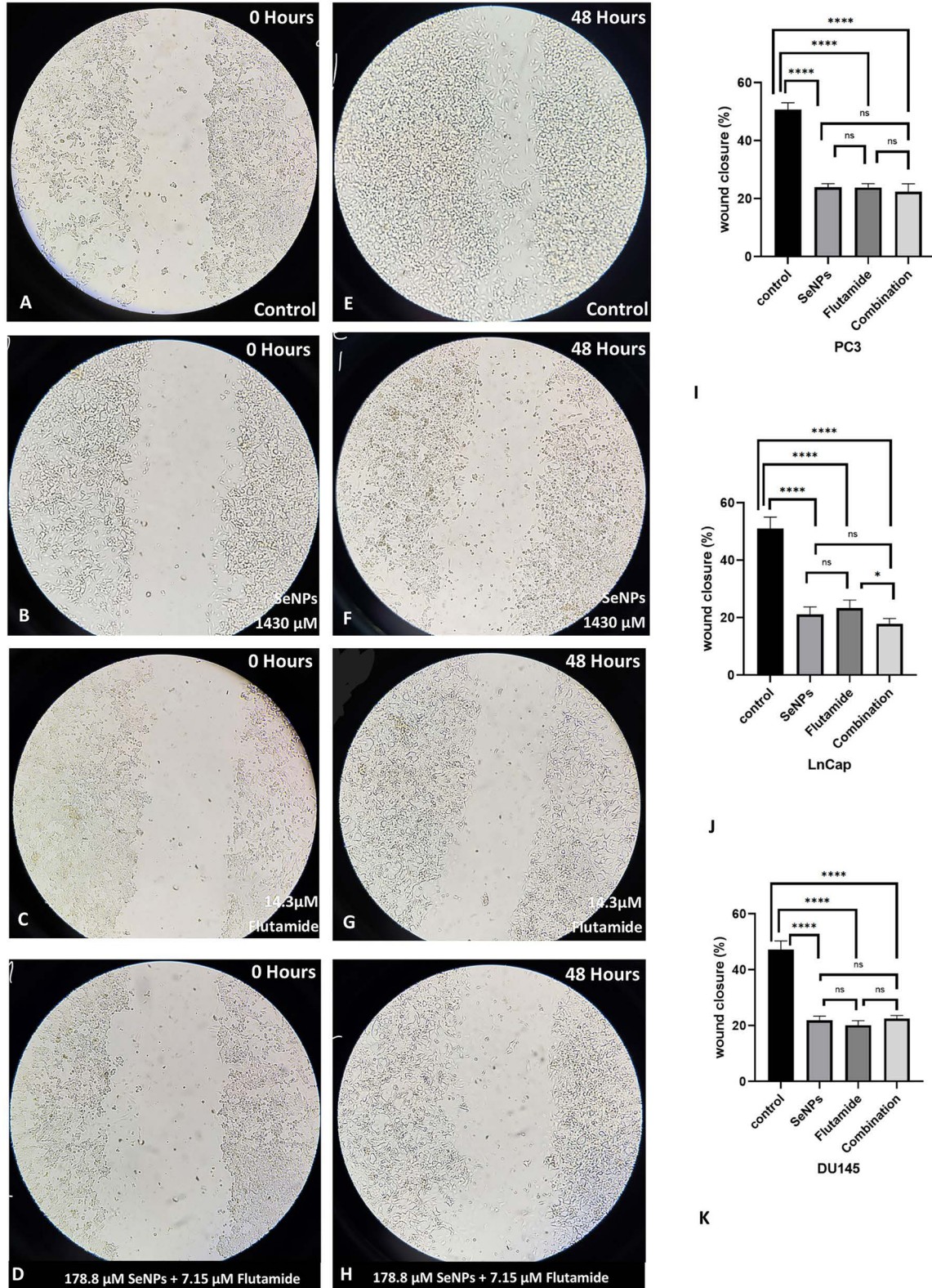

**Fig 5. Migration assay for LnCap prostate cancer cells.** (A) control-Day 0 (B) SeNPs-Day 0 (C) Flutamide-Day 0 (D) Combination-Day 0 (E) control-Day 2 (F) SeNPs-Day 2 (G) Flutamide-Day 2 (H) combination-Day 2 (I–K) all treated group significantly inhibited movements of prostate cancer cell lines into the wound after 48 hours. Statistical significance was defined at *p < 0.05, **p < 0.01, ***p < 0.001, and ****p < 0.0001.

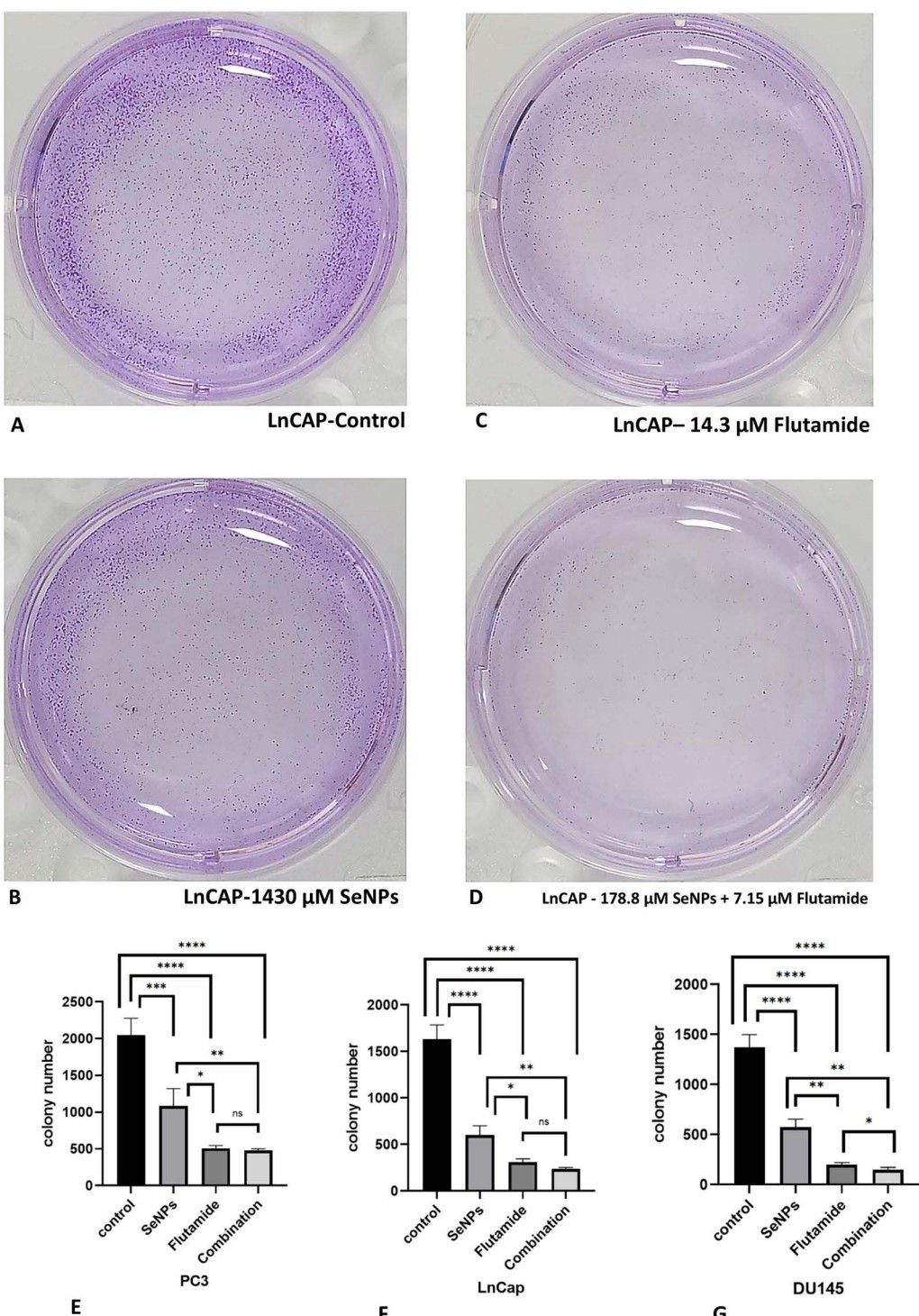

**Fig 6. Colony formation assay in LNCaP prostate cancer cells.** (A) Control (B) SeNPs (C) Flutamide (D) combination. (E-G) SeNPs, Flutamide, and their combinations in IC50 concentrations prompted a marked reduction in colony count across each one of the stated prostate cancer cell lines. Statistical significance was defined at *p < 0.05, **p < 0.01, ***p < 0.001, and ****p < 0.0001.

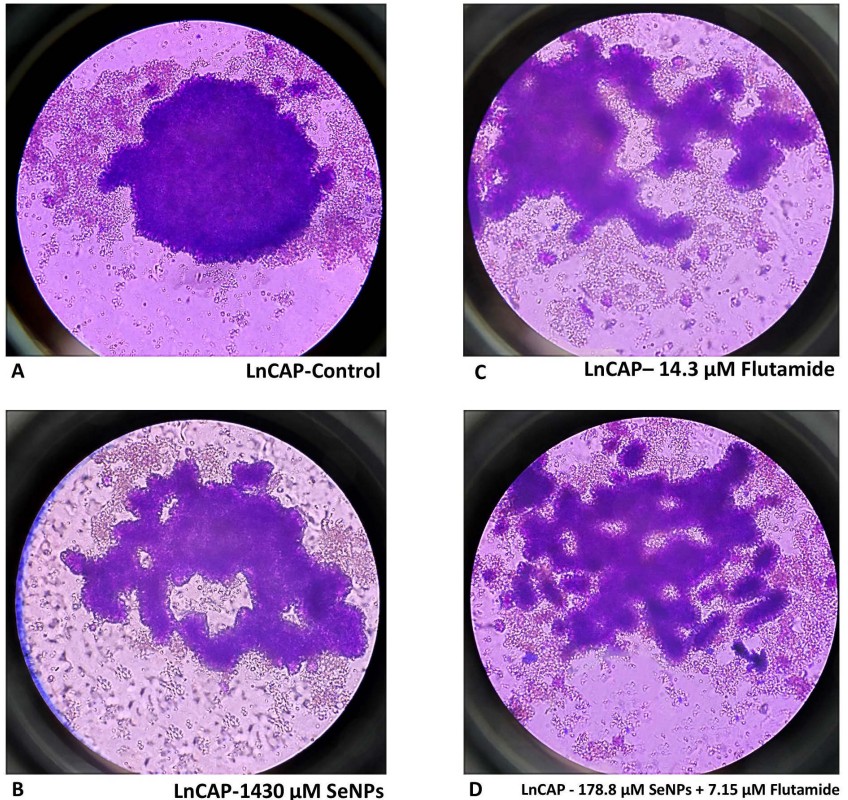

**Fig 7. LnCAP tumor spheroid model.** Treated spheroids showed notable structural damage and a high presence of dead cells. In contrast, the untreated spheroids maintained a typical intact structure. (A) control (B) SeNPs (C) Flutamide (D) combination.

### 3.9. SeNPs and Flutamide induce SubG1-G1 arrest in the prostate cancer cell lines

The flow cytometric assay of the prostate cancer cell lines PC3, LNCaP, and Du145 subjected to SeNPs, Flutamide, and their combinations in relation to the induction of cell cycle arrest is illustrated in Fig 9. Cells in the phases sub-G1, G1, S, and G2/M phases accounted for 5.79%, 44.4%, 34.8%, and 14.4% of the total cell sample in the control group of PC3 metastatic prostate cancer cells.

When PC3 metastatic prostate cancer cells were treated with SeNPs, there was an increase in sub-G1 cells (5.79% to 60.3%, $p < 0.0001$) and a decrease in G1 (44.4% to 12.3%, $p < 0.0001$), S (34.8% to 30.3%, $p < 0.05$) and G2/M phase (14.4% to 0.53%, $p < 0.001$) cells. Treatment with Flutamide also caused an increase in sub-G1 cells (5.79% to 28.6%, $p < 0.0001$), and S (34.8% to 44.0%, $p < 0.05$) cells, while decreasing G1 (44.4% to 20.3%, $p < 0.0001$), and G2/M (14.4% to 5.94%, $p < 0.001$) phases. The combination therapy of Flutamide plus SeNPs resulted in a significant increase in sub-G1 cells (5.79% to 48.7%, $p < 0.0001$) for the PC3 cell line. Similar results were observed for LNCaP prostate cancer cells, with an increase in sub-G1 phase cells following treatment with SeNPs (4.61% to 31.8%, $p < 0.0001$), Flutamide (4.61% to 51.6%, $p < 0.0001$), and their combination (4.61% to 29.7%, $p < 0.0001$).

In summary, the outcome of the current study indicates that the administration of SeNPs, Flutamide, and their combination effectively halted the progression of the cell cycle in the SubG1/G1 stage, compared to the control and untreated subgroups. Furthermore, the

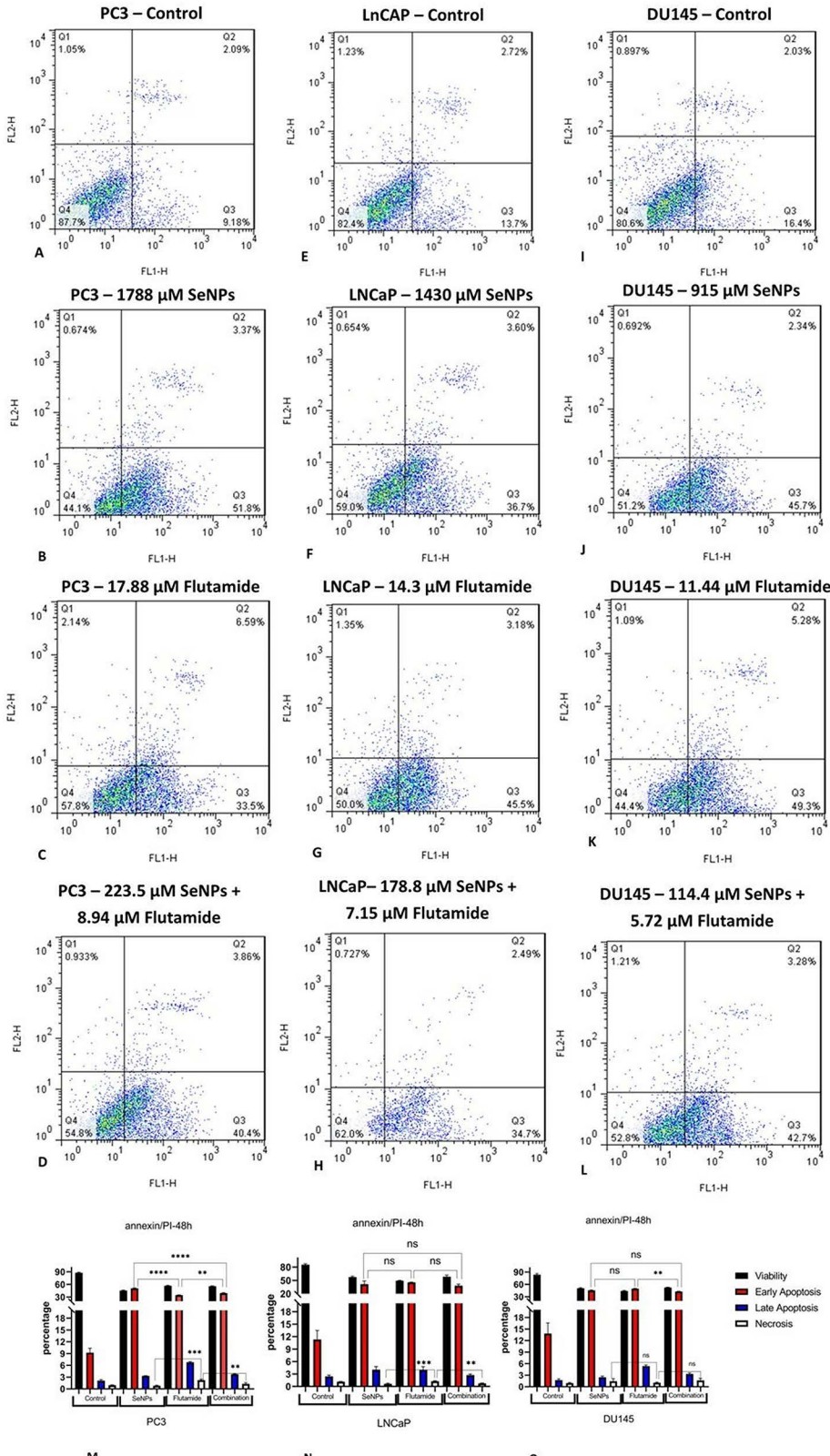

**Fig 8. Flow cytometric examination of PC3, LnCAP, and DU145 cell apoptosis using Annexin-V-Flous after 48 hours.** (A–D) Induced apoptosis of PC3 cell untreated and treatment with SeNPs, Flutamide, and their combination

(In order from up to down). (E–H) Induced apoptosis of LNCaP cell untreated and treatment with SeNPs, Flutamide, and their combination (In order from up to down). (I–L) Induced apoptosis of DU145 cell untreated and treatment with SeNPs, Flutamide, and their combination (In order from up to down). (M-O) there was a significant enhancement in apoptosis of all treated groups compared to the control group in PC3, LnCAP, and DU145 cell lines. Statistical significance was defined at *p < 0.05, **p < 0.01, ***p < 0.001, and ****p < 0.0001.

presence of the sub-G1 peak in the flow cytometry graphs suggests apoptosis in the cancer cells (Fig 9).

## 3.10. Effect of Flutamide and SeNPs on gene expression levels in prostate cancer cell line

All three prostate cancer cell lines underwent treatment with IC50 doses of SeNPs, Flutamide, and their combination for a duration of 48 hours. Following this, the expression levels of apoptosis-related genes (*BAX*, *Bcl-2*), an angiogenesis gene (*VEGF-C*), a gene associated with progression and development (*HIF-1α*), genes involved in the EMT pathway (*SNAIL* and *E-Cadherin*), and a prostate cancer marker (*KLK3*) were evaluated using Real-Time PCR. The resulting data was analyzed with the help of GraphPad Prism version 9 software, as shown in Fig 10.

As depicted in Fig 10, in the LnCAP and DU145 cell lines, all treatment groups, whether individual or in combination, led to a significant decrease in the expression of *VEGF-C* and *HIF-1α* genes (p < 0.05). Regarding the EMT pathway genes, each group that received treatment saw a significant increase in *E-cadherin* expression and a decrease in *SNAIL* gene expression (p < 0.05). Furthermore, all medication groups significantly reduced the expression of the prostate cancer biomarker known as *KLK3* (p < 0.05). Additionally, every group that underwent treatment up-regulated *BAX* gene expression significantly and down-regulated *Bcl-2* gene expression. Thus, it is suggested that SeNPs could act as a promising anti-cancer treatment for LnCAP and DU145 cells through the induction of apoptosis via the mitochondrial pathway.

In the PC3 cell line, all treatment groups led to a significant decrease in *VEGF-C* expression (p < 0.05). Flutamide and the combination of Flutamide with SeNPs significantly decreased *HIF-1α* gene expression, while SeNPs alone showed a non-significant decrease. In terms of EMT pathway genes, every group that was treated saw a significant increase in *E-cadherin* expression (p < 0.05), but the decrease in *SNAIL* gene expression was not significant in any treatment groups. All treatment groups reduced the expression of the prostate cancer biomarker *KLK3*, with only the flutamide group showing a significant change (p < 0.05). Moreover, all treated groups significantly enhanced *BAX* gene expression and decreased *Bcl-2* gene expression. By inducing apoptosis via the mitochondrial route, SeNPs could consequently function as a prospective anti-cancer agent toward PC3 cells.

These results suggest that the IC50 concentration of SeNPs and Flutamide combination (223.5 μM SeNPs + 8.94 μM Flutamide for PC3, 178.8 μM SeNPs + 7.15 μM Flutamide for LnCAP, and 114.4 μM SeNPs + 5.72 μM Flutamide for DU145) used in this Real-Time PCR test had similar effects to the IC50 concentration of Flutamide alone (17.88 μM, 14.3 μM, and 11.44 μM for PC3, LnCAP, and DU145 cell lines, respectively). The changes in gene expression mentioned were more significant in LnCAP than in DU145 cell lines, and in DU145 more than in PC3 cell lines.

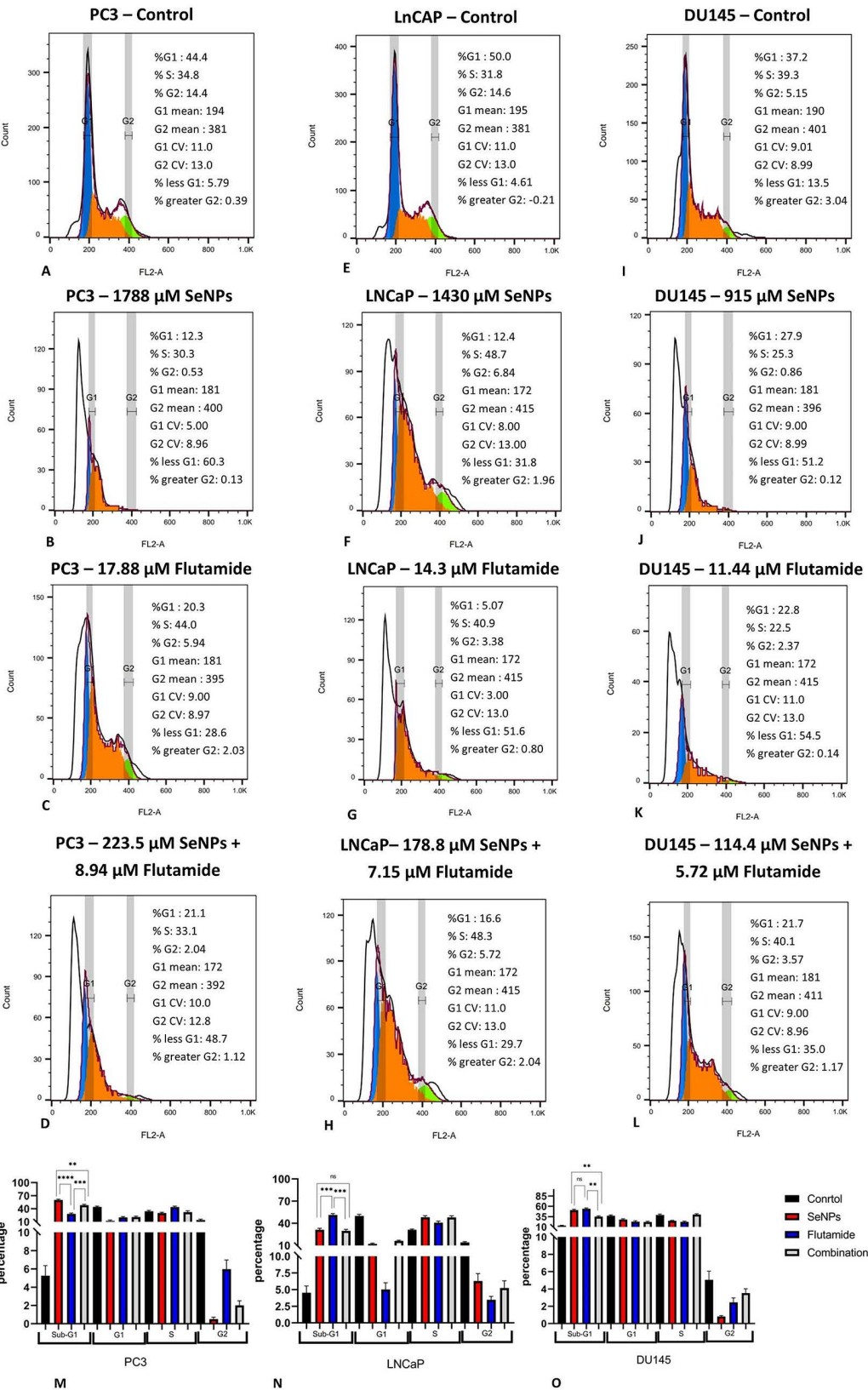

**Fig 9. Cell cycle assessment of PC3, LnCAP, and DU145 cell lines.** (A–D) Cell cycle analysis of PC3 cell untreated and treatment with SeNPs, Flutamide, and their combination (In order from up to down). (E–H) Cell cycle analysis of

LNCaP cell untreated and treatment with SeNPs, Flutamide, and their combination (In order from up to down). (I–L) Cell cycle analysis of DU145 cell untreated and treatment with SeNPs, Flutamide, and their combination (In order from up to down). (M–O) According to the cell cycle analysis, we detected sub-G1 area increased significantly in PC3, LnCAP, and DU145 cells when treated with SeNPs, Flutamide, and their combination. Statistical significance was defined at *p < 0.05, **p < 0.01, ***p < 0.001, and ****p < 0.0001.

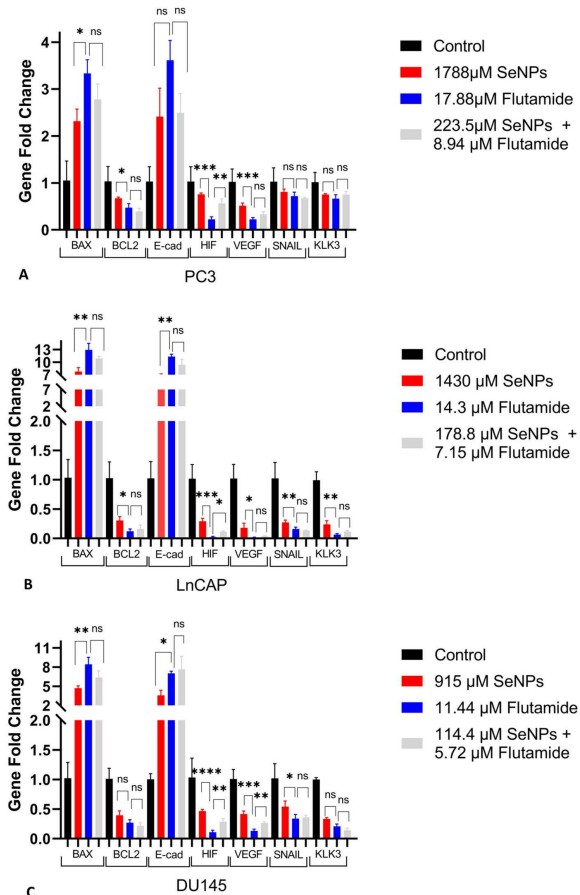

**Fig 10. The gene expression rate results of three prostate cancer cell lines.** PC3 (A), LnCAP (B), DU145 (C) treated with SeNPs, Flutamide, their combination on the expression of target genes for 48 h. Values are given as mean ± SD of three independent experiments. Statistical significance was defined at *p < 0.05, **p < 0.01, ***p < 0.001, and ****p < 0.0001.

## 4. Discussion

When it comes to treating developed and metastatic prostate cancer, hormonal treatment remains the main curative option [30]; however, certain individuals develop resistance to ADT and progress to CRPC, with only a small percentage of them showing a long-term response [31]. Drug resistance, expenses, inaccessibility, and serious negative consequences such as hepatotoxicity restrict their use in clinical settings. For instance, flutamide is known to induce liver damage [19]. Due to unsatisfactory monotherapy outcomes in CRPC clients, there has been a significant focus on conjunction treatment, which has demonstrated greater efficacy. Numerous investigations have shown the advantages of combining chemotherapy

with selenium, among them using selenocystine with 5-fluorouracil, auranofin with doxorubicin, and methylseleninic acid with cisplatin [32–35].

The combination of flutamide and SeNPs for the therapy of metastatic PCa has not been evaluated in any study to date. Our findings show that there is a synergistic reduction in LnCAP, PC3, and DU145 cell survival when flutamide is combined with SeNPs. This combination also promotes apoptosis and inhibits cell growth by inducing cell cycle arrest. Additionally, we observed a decrease in the expression of *KLK3*, angiogenesis and proliferation genes (*VEGF-C* and *HIF-1α*), the apoptosis gene (*Bcl-2*), and *SNAIL* in all cell lines treated with SeNPs, Flutamide, or both compared to the control group. On the other hand, the expression of *E-cadherin* and *BAX* genes increased. The expression of the mentioned genes (except *HIF-1α*) in the combination group and flutamide group had no significant difference (p > 0.05) in the LnCAP cell line. Therefore, SeNPs have the potential to be highly beneficial in treating metastatic PCa and should be considered as a powerful therapy option for this condition, taking into account both its advantages and negative effects. Combining flutamide with SeNPs may reduce the required intake of flutamide and mitigate its negative effects, particularly its hepatotoxicity.

Multiple non-randomized tests have suggested that flutamide could be an additional treatment option in hormone therapy [36–38]. However, the precise mechanisms responsible for the anti-tumor effects of flutamide are not yet fully understood. Shen and colleagues discovered that flutamide enhances cell death and inhibits cell growth, possibly through the upregulation of *KLF9* [39]. Consistent with our findings, canine *KLK2* values fell in dogs that have been administered flutamide, as reported by Mitchell G. Lawrence et al. [40].

Inorganic, organic, and SeNPs are only a few of the many selenium classes that are now accessible for employing in anti-cancer protocols. Various forms of selenium compounds have distinct biological effects due to their differing physical characteristics, chemical compositions, metabolic pathways in cells and tissues, and the range of molecules they impact within cells. SeNPs are nanoparticles that provide distinct benefits such as minimal toxicity and rapid uptake by cells compared to other forms of selenium [41,42]. SeNPs have gained attention for their superior anti-cancer properties and lower toxicity in contrast to other inorganic and organic selenium compounds, making them a promising candidate for cancer prevention [43–46]. The anticancer action of selenium is believed to be attributed to various mechanisms, including cell cycle arrest, antioxidation, apoptosis, and disruption of cell signaling processes [7,8,16,47–49].

Sonkusre and colleagues demonstrated that SeNPs at a dosage of 2 μg Se/ml induced necroptosis in PC-3 cells through internalization, driven by reactive oxygen species (ROS). The real-time qPCR analysis indicated an upregulation in the expression of necroptosis-associated tumor necrotic factor (*TNF*) and interferon regulatory factor 1 (*IRF1*). SeNPs administration led to higher *RIP1* protein expression at the translational phase. Additionally, cell survival was notably enhanced when treated with the necroptosis inhibitor, Necrostatin-1 [50]. Biogenic selenium nanoparticles produced from Bacillus licheniformis may cause necroptosis in LNCaP-FGC cells at a dosage of 2 μg Se/ml, without impacting the integrity of red blood cells. Real-time gene expression estimation showed increased levels of *TNF* and *IRF1*, and reduced levels of androgen receptor (AR) and PSA [51]. The dosage of SeNPs used in our study was higher than in the studies mentioned. Additionally, levels of the androgen receptor were not evaluated in our study.

Yulin An alongside his team explored the therapeutic effects of lentinan-functionalized selenium nanoparticles (LET-SeNPs) combined with zoledronic acid (ZOL) on metastatic PCa cells. The study demonstrated that this combination could significantly reduce the growth of PCa cells in a concentration-dependent manner, as shown through cytotoxicity assays, flow

cytometry, and tests on mitochondrial functions. The enhanced anticancer properties of LET-SeNPs with ZOL were linked to the modulation of BCL2 family proteins, which promotes the release of cytochrome C from the mitochondria into the cytoplasm. This action triggers cell cycle arrest in the S phase, leading to irreversible DNA damage and the eradication of PCa cells. Consequently, the research highlights that SeNPs and ZOL together offer a potent solution for curtailing the expansion of PCa cells [52].

There is a limited number of animal research studies investigating the anticancer effects of SeNPs. Sonkusre's histopathology research demonstrated that ingesting selenium nanoparticles at a concentration of 50 mg Se/kg of body weight resulted in significantly decreased toxicity compared to L-selenomethionine (5 mg Se/kg) [51]. Furthermore, Shahverdi et al. discovered that folic acid surface-coated selenium nanoparticles exhibited in vivo benefits for breast cancer. They found that SeNPs could inhibit PCa growth by reducing *IKK-ε* and *SMAD2* mRNA levels in xenograft mice, suggesting that SeNPs could be used for anti-tumor purposes. The effectiveness of SeNPs dropped notably in mice treated with miR-155-5p shRNA, confirming the crucial role of miR-155-5p in facilitating SeNPs' anti-tumor properties in living creatures [53].

The precise biochemical process that enables SeNPs to trigger tumor inhibition remains incompletely understood. It is widely accepted that SeNPs can induce tumor cell death by increasing cellular absorption and inhibiting ROS [54]. Huang and colleagues have shown that selenium nanoparticles may induce autophagy in cancer cells, therefore exhibiting an anticancer effect [55]. Sonkusre's findings revealed that SeNPs led to an increase in TNF, ultimately resulting in the activation of cancer cell necrosis [50]. Vekariya et al. found that SeNPs impeded cell development and the synthesis of DNA, RNA, and proteins, indicating that SeNPs could impact the regulation of various relevant molecules, including non-coding RNA [56].

In order to assess the potential pathway of apoptosis triggered by SeNPs, we examined the expression of the apoptotic gene *BAX* and the anti-apoptotic gene *Bcl-2*. The ratio of these genes was considered indicative of the drug's impact on mitochondrial apoptosis. After the administration of SeNPs, Flutamide, or a combination of both, there was a decrease in the expression of the anti-apoptotic *Bcl-2* gene and an increase in the expression of the pro-apoptotic *BAX* gene. This shift in gene expression led to increased apoptosis, which was linked to the expression of the cyclin-related inhibitor p21 [57]. *Bcl-2* family proteins play a role in regulating programmed cell death by influencing mitochondrial function [58]. Indeed, depending on many physiological and pathological conditions, alterations in the electrical potential of mitochondria may result in a discharge of apoptotic proteins, including pro-caspases 2, 3, and 9 [59].

It is hypothesized that the production of ROS is essential for mitochondrial oxidative damage, which triggers apoptosis [60]. By reducing cellular ROS, *Bcl-2* performs an anti-apoptotic action in this way [59]. As a consequence, intracellular ROS buildup that leads to mitochondrial-mediated apoptosis is favored by downregulating *Bcl-2* and overexpressing *BAX*. Nevertheless, the P53 protein, a tumor suppressor that triggers cell cycle arrest and death in response to damage to the genome, has two well-known transcriptional targets: *BAX* and *Bcl-2* [61]. Taken together, The enhanced expression of *BAX* and diminished expression of *Bcl-2* support apoptosis and reduced growth of PCa cells.

We investigated two crucial regulators of angiogenesis, *VEGF-C*, and *HIF-1α*, along with genes related to apoptosis. *VEGF-C* is a powerful pro-angiogenic agent that plays a crucial role in microvascular restructuring, angiogenesis, and the advancement of PCa spread and progression [62]. *VEGF-C* expression has been linked to the advancement of prostate cancer and lymph node metastasis, according to studies done by Yang and Jenbacken [63,64].

Our findings suggest that the androgen communication channel may be involved in the SeNPs-induced changes in the expression of *VEGF-C* and *HIF-1α* genes. Treating androgen-dependent LNCaP cells with SeNPs resulted in a significant drop in *VEGF-C* and *HIF-1α* expression. In androgen-independent PC-3 cells, SeNPs treatment also reduced the expressions of *VEGF-C* and *HIF-1α*, but to a lesser extent compared to LnCAP cells. These results suggest that SeNPs may be beneficial for prostate cancer treatment, particularly in the early stages. Further extensive studies is required to validate and support our findings.

Recent investigations have shown that human kallikrein-related peptidase 3 (*KLK3*), a protease found exclusively in the prostate, can promote the advancement of prostate cancer and serve as a biomarker [65,66]. Studies have shown that increased expression of the *KLK3* gene in prostate cancer is linked to several processes such as cell growth, movement, infiltration, blood vessel formation, and resistance to cell death [67,68]. *KLK3* has been utilized in several research studies as a biological indicator for prostate cancer prognosis and as a target for therapy. Treatment with SeNPs, Flutamide, or a combination of them significantly reduced *KLK3* mRNA expression in all prostate cancer cell lines, confirming the tumor-inhibiting properties of these chemicals.

We suggest that future studies delve deeper into specific issues. They could focus on studying the impact of SeNPs on cancer stem cells, which play a crucial role in the recurrence of the disease due to their resistant characteristics. Additionally, our findings can be validated through knockdown/overexpression studies employing RNAi and viral-based gene therapies. Subsequent research could involve using xenograft murine models to evaluate the adaptive response of prostate cancer cells to SeNPs in vivo. The Western blot technique may be utilized to examine the protein expression of *VEGF-C*, *KLK2*, and the EMT system.

## 5. Conclusion

We have demonstrated for the first time that the combination therapy of flutamide and SeNPs enhances the anti-tumor effect on PCa cell lines, especially in LnCAP cells. This is due to their ability to induce apoptosis and prevent the growth of PCa cells. It is suggested that the concurrent administration of SeNPs and flutamide could potentially reduce the effective dosage of flutamide and mitigate its adverse effects, particularly hepatotoxicity.

## Supporting information

**S1 Fig. Migration assay for Du145 prostate cancer cells.** (A) control-Day 0 (B) SeNPs-Day 0 (C) Flutamide-Day 0 (D) Combination-Day 0 (E) control-Day 2 (F) SeNPs-Day 2 (G) Flutamide-Day 2 (H) combination-Day 2.
(TIF)

**S2 Fig. Migration assay for PC3 prostate cancer cells.** (A) control-Day 0 (B) SeNPs-Day 0 (C) Flutamide-Day 0 (D) Combination-Day 0 (E) control-Day 2 (F) SeNPs-Day 2 (G) Flutamide-Day 2 (H) combination-Day 2.
(TIF)

**S1 Raw Image. The original unprocessed images captured during the experiments, presented without any modifications or enhancements.** These images serve as direct visual representations of the experimental results.
(PDF)

## Author contributions

**Conceptualization:** Seyed Mohammad Kazem Aghamir.

**Formal analysis:** Ramin Heshmat.

**Methodology:** Iman Menbari Oskouie, Akram Mirzaei, Seyedeh Negin Hashemi Dougaheh, Helia Azodian Ghajar.

**Resources:** Rahil Mashhadi.

**Validation:** Fatemeh Khatami, Amin Shiralizadeh Dezfuli, Rahil Mashhadi, Akram Mirzaei, Seyedeh Negin Hashemi Dougaheh, Helia Azodian Ghajar.

**Writing – original draft:** Iman Menbari Oskouie.

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
