## [Decision Letter · Decision Letter 0]

30 Oct 2024

PONE-D-24-41345Reducing the Effective Dosage of Flutamide on Prostate Cancer Cell Lines through Combination with Selenium Nanoparticles: An In-Vitro StudyPLOS ONE

Dear Dr. aghamir,

Thank you for submitting your manuscript to PLOS ONE. After careful consideration, we feel that it has merit but does not fully meet PLOS ONE’s publication criteria as it currently stands. Therefore, we invite you to submit a revised version of the manuscript that addresses the points raised during the review process.

We look forward to receiving your revised manuscript.

Kind regards,

Ahmed E. Abdel Moneim

Academic Editor

PLOS ONE

Journal Requirements:

4. PLOS ONE now requires that authors provide the original uncropped and unadjusted images underlying all blot or gel results reported in a submission’s figures or Supporting Information files. This policy and the journal’s other requirements for blot/gel reporting and figure preparation are described in detail at https://journals.plos.org/plosone/s/figures#loc-blot-and-gel-reporting-requirements and https://journals.plos.org/plosone/s/figures#loc-preparing-figures-from-image-files. When you submit your revised manuscript, please ensure that your figures adhere fully to these guidelines and provide the original underlying images for all blot or gel data reported in your submission. See the following link for instructions on providing the original image data: https://journals.plos.org/plosone/s/figures#loc-original-images-for-blots-and-gels. In your cover letter, please note whether your blot/gel image data are in Supporting Information or posted at a public data repository, provide the repository URL if relevant, and provide specific details as to which raw blot/gel images, if any, are not available. Email us at plosone@plos.org if you have any questions.

Reviewers' comments:

Reviewer's Responses to Questions

**Comments to the Author**

1. Is the manuscript technically sound, and do the data support the conclusions?

Reviewer #1: Yes

Reviewer #2: Yes

Reviewer #3: Yes

Reviewer #4: Yes

2. Has the statistical analysis been performed appropriately and rigorously? 

Reviewer #1: I Don't Know

Reviewer #2: Yes

Reviewer #3: Yes

Reviewer #4: N/A

3. Have the authors made all data underlying the findings in their manuscript fully available?

Reviewer #1: Yes

Reviewer #2: Yes

Reviewer #3: Yes

Reviewer #4: Yes

4. Is the manuscript presented in an intelligible fashion and written in standard English?

Reviewer #1: Yes

Reviewer #2: Yes

Reviewer #3: No

Reviewer #4: No

5. Review Comments to the Author

Reviewer #1: 1- Cell viability, gene expression and annexin PI data showed obviously that SeNPs didn't reduce IC50 Of Flutamide

2- In figure 4, you may need to upload better images, background look like hazy images.

3- In figure 9, LANCAP and DU145 control cells showed highly DNA damage, Would you mind explain why some cells have 35% Sub-G1?

4- Western blots measure the above plus translation efficiency, transport, and protein stability and perhaps post translational modifications. I would recommend to perform WB instead of real-time PCR.

Reviewer #2: Well done! This manuscript is good

1. Please make all required changes.

2. Some pictures need clarification.

4. format your references

5. Please format the references and add the name of the journal and Vol.

Reviewer #3: Dear Authors,

The manuscript 'Reducing the Effective Dosage of Flutamide on Prostate Cancer Cell Lines

through Combination with Selenium Nanoparticles: An In-Vitro Study' is interesting, But many grammatical errors I have found in your paper, Then, you can fix them all.

Reviewer #4: Dear colleagues

The work is in a good state but I think the below comments need to be modified which will enhance the quality of the manuscript.

-Some editing for English language is required throughout the manuscript

-It is suggested to extract and use the latest cancer incidence and mortality statistics, especially the cancers investigated in this research, from the GLOBACA website.

-All the genes must be italic. In scientific names, the genus name begins with a capital letter.

-Add clearly the hypothesis, aims and goals of this work to the last paragraph to your introduction.

-In material &method don’t have enough reference. This article could be substantially improved in this case:

https://doi.org/10.1007/s13204-022-02386-w

https://doi.org/10.3390/inorganics11080322

https://doi.org/10.1007/s12011-020-02054-6

-Authors mentioned SeNPs have a diameter of around 45.34 nm and are spherical, as demonstrated by the TEM image. Whereas in the TEM image there is no size mentioned. Only scale is mentioned in the image Fig. 1A.

-Figure 2 :MTT assay -Include number of repeats/observations per assay point and also include statical analysis.

-In figure 4, the changes made in nanoparticle-treated cells are shown using arrows on the figure.

-Why is the combination of SeNPs and Flutamide not explained? What advantage do you expect for this combination? There is still a lack of relevant experimental evidence for the mechanistic explanation of the mechanism.

-How do you justify your study provides a promising approach to cancer treatment?.

6. PLOS authors have the option to publish the peer review history of their article (what does this mean? ). If published, this will include your full peer review and any attached files.

**Do you want your identity to be public for this peer review?** For information about this choice, including consent withdrawal, please see our Privacy Policy .

Reviewer #1: **Yes: ** Mostafa A.L AbdelSalam

Reviewer #2: **Yes: ** Shimaa Elsayed Rashad

Reviewer #3: No

Reviewer #4: No

---

## [Author Response · Author response to Decision Letter 1]

8 Dec 2024

Dear Editor-in-Chief

We appreciate your insightful feedback on our manuscript. After thoroughly reviewing your comments, we have implemented the necessary adjustments in the revised version. Below, you will find the responses to your comments.

1. Please ensure that your manuscript meets PLOS ONE's style requirements, including those for file naming. The PLOS ONE style templates can be found at https://journals.plos.org/plosone/s/file?id=wjVg/PLOSOne_formatting_sample_main_body.pdf and MailScanner has detected a possible fraud attempt from "track.editorialmanager.com" claiming to be https://journals.plos.org/plosone/s/file?id=ba62/PLOSOne_formatting_sample_title_authors_affiliations.pdf

Re: Thank you for your feedback and for highlighting the need to ensure compliance with PLOS ONE’s style requirements, including file naming conventions. We have made the necessary revisions.

Re: We have provided a complete Data Availability Statement.

Re: We have included the full ethics statement in the ‘Methods’ section. The study was conducted at the Urology Research Center, Tehran University of Medical Sciences, following the receipt of REB authorization from the Ethics Review Board of Sina Hospital.

4. PLOS ONE now requires that authors provide the original uncropped and unadjusted images underlying all blot or gel results reported in a submission’s figures or Supporting Information files. This policy and the journal’s other requirements for blot/gel reporting and figure preparation are described in detail at https://journals.plos.org/plosone/s/figures#loc-blot-and-gel-reporting-requirements and https://journals.plos.org/plosone/s/figures#loc-preparing-figures-from-image-files. When you submit your revised manuscript, please ensure that your figures adhere fully to these guidelines and provide the original underlying images for all blot or gel data reported in your submission. See the following link for instructions on providing the original image data: https://journals.plos.org/plosone/s/figures#loc-original-images-for-blots-and-gels. In your cover letter, please note whether your blot/gel image data are in Supporting Information or posted at a public data repository, provide the repository URL if relevant, and provide specific details as to which raw blot/gel images, if any, are not available. Email us at plosone@plos.org if you have any questions.

Re: We have provided the original figures.

Review Comments to the Author

Reviewer #1:

1- Cell viability, gene expression and annexin PI data showed obviously that SeNPs didn't reduce IC50 Of Flutamide

Re: Thank you for your comments regarding the effects of SeNPs on the IC50 of Flutamide.

Regarding cell viability, our MTT assay results indicate that the IC50 values after 48 hours of treatment with Flutamide alone were 17.88 μM, 14.3 μM, and 11.44 μM for PC3, LNCaP, and DU145 cell lines, respectively. However, when combined with SeNPs, which allowed for a 50% reduction in the Flutamide IC50, the new IC50 values for the combination therapy were 223.5 μM SeNPs + 8.94 μM Flutamide for PC3, 178.8 μM SeNPs + 7.15 μM Flutamide for LNCaP, and 114.4 μM SeNPs + 5.72 μM Flutamide for DU145. These results demonstrate that by combining Flutamide with SeNPs, the effective dosage of Flutamide could be significantly reduced, exemplified by the reduction from 17.88 μM to 8.94 μM in the PC3 cell line with the addition of 223.5 μM SeNPs.

In terms of gene expression analysis via real-time PCR and the annexin PI assay, our comparisons between the Flutamide alone and the combination treatment groups suggest similar outcomes. Importantly, these similar results were achieved with a halved concentration of Flutamide when combined with an appropriate concentration of SeNPs. This suggests that SeNPs may facilitate the reduction of the effective dosage of Flutamide while maintaining efficacy.

We appreciate your review and hope this addresses your concerns effectively.

2- In figure 4, you may need to upload better images, background look like hazy images.

Re: We appreciate your suggestion to improve the quality of Figure 4. We have carefully reviewed the figure and agree that the image was not as clear as it could be. Therefore, we have replaced the original image with a higher-resolution version that addresses the issue of the hazy background.

3- In figure 9, LANCAP and DU145 control cells showed highly DNA damage, Would you mind explain why some cells have 35% Sub-G1?

Re: Thank you for your insight. We identified some errors in our cell cycle assay experiment and analysis. As a result, we have repeated the experiment for all three cell lines and treatment groups. The updated results are presented in Figure 9, and the relevant sections have been revised accordingly.

4- Western blots measure the above plus translation efficiency, transport, and protein stability and perhaps post translational modifications. I would recommend to perform WB instead of real-time PCR.

Re: However, we are currently facing significant resource constraints at our institute, which impact our ability to perform additional experimental procedures such as Western blotting. Our research operates under a tightly controlled budget, and at this time, we must prioritize the methodologies we can conduct within our existing financial framework. As our funding becomes more robust, we aim to incorporate additional techniques like Western blotting in future projects.

In the meantime, we have employed real-time PCR to effectively measure gene expression levels. While we recognize its limitations compared to Western blotting for the purposes you mentioned, we believe it provides reliable and relevant insights for our current study.

We appreciate your understanding and hope that our explanation clarifies the constraints we are working under. We are grateful for your constructive feedback and remain committed to advancing our research with the resources available to us.

Reviewer #2:

Well done! This manuscript is good

1. Please make all required changes.

Re: Thank you for taking the time to review our manuscript. The required changes have been made.

2. Some pictures need clarification.

Re: Thank you for your suggestion. We have reviewed all the figures and enhanced the clarity of the images that needed improvement. Adjustments have been made to ensure they are clear and easy to interpret.

3. format your references

Re: We have changed the format of our references

4. Please format the references and add the name of the journal and Vol.

Re: We have reformatted the reference section to include all necessary details, including the journal names and volume numbers, in line with the journal’s guidelines

Reviewer #3:

Dear Authors,

The manuscript 'Reducing the Effective Dosage of Flutamide on Prostate Cancer Cell Lines

through Combination with Selenium Nanoparticles: An In-Vitro Study' is interesting, But many grammatical errors I have found in your paper, Then, you can fix them all.

Re: Thank you for your careful reading of our manuscript and for highlighting the importance of addressing grammatical issues. We have thoroughly reviewed the manuscript, and corrected the grammatical errors throughout the text.

We believe these revisions enhance the clarity and readability of our work. If there are specific sections that you feel require additional attention, please let us know, and we will address them promptly.

We appreciate your valuable feedback and the opportunity to improve our manuscript.

Reviewer #4:

Dear colleagues

The work is in a good state but I think the below comments need to be modified which will enhance the quality of the manuscript.

-Some editing for English language is required throughout the manuscript

Re: Thank you for your valuable feedback and for pointing out the need for English language editing. We have conducted a comprehensive revision of the manuscript to address language issues and ensure clarity and coherence throughout the text.

-It is suggested to extract and use the latest cancer incidence and mortality statistics, especially the cancers investigated in this research, from the GLOBACA website.

Re: Thank you for your suggestion to include the most recent cancer incidence and mortality statistics. We have updated the manuscript with the latest data from the GLOBOCAN website, focusing on prostate cancer, which is the primary subject of our research.

-All the genes must be italic. In scientific names, the genus name begins with a capital letter.

Re: We have revised the manuscript to ensure that all gene names are italicized for consistency and clarity. Additionally, scientific names have been formatted correctly, with genus names beginning with a capital letter and species names in lowercase.

-Add clearly the hypothesis, aims and goals of this work to the last paragraph to your introduction.

Re: We have included the hypothesis, aims and goals of this work in the last paragraph of the introduction

-In material &method don’t have enough reference. This article could be substantially improved in this case:

https://doi.org/10.1007/s13204-022-02386-w

https://doi.org/10.3390/inorganics11080322

https://doi.org/10.1007/s12011-020-02054-6

Re: Thank you for your suggested references. We have added them to the method section.

-Authors mentioned SeNPs have a diameter of around 45.34 nm and are spherical, as demonstrated by the TEM image. Whereas in the TEM image there is no size mentioned. Only scale is mentioned in the image Fig. 1A.

Re: Regarding the diameter of SeNPs in the TEM image (45.34 nm), it was calculated using ImageJ software. The diameter of a twenty random particles was measured, and the average diameter was then calculated.

-Figure 2 :MTT assay -Include number of repeats/observations per assay point and also include statical analysis.

Re: Thank you for your valuable and perceptive feedback. We have included the number of repeats/observations in the methods section. Additionally, we have included the statistical analysis (Figure 2).

-In figure 4, the changes made in nanoparticle-treated cells are shown using arrows on the figure.

Re: We have added arrows to cells that have been changed after treatment with SeNPs and Flutamide

-Why is the combination of SeNPs and Flutamide not explained? What advantage do you expect for this combination? There is still a lack of relevant experimental evidence for the mechanistic explanation of the mechanism.

Re: Flutamide is a non-steroidal antiandrogen drug that primarily inhibits the effects of androgens, such as testosterone, at the cellular level. This characteristic makes it valuable in treating a range of medical conditions associated with excessive androgen activity, including prostate cancer and certain hormonal disorders. Flutamide functions by competing with dihydrotestosterone for binding to the androgen receptor (AR) (1-4). Notably, flutamide has been documented to act as an AR agonist in human prostate cancer tissues with AR gene mutations and in LNCaP cells (5-7). Since PC3 cells are known to lack AR, it was anticipated that flutamide would not significantly affect them. However, in line with our findings, Jong-Jae observed that flutamide alters the expression of certain genes in PC3 cells (8), suggesting that flutamide might influence gene expression through alternative pathways.

Protein kinase C (PKC) isoenzymes, including PKC-α, PKC-β1, PKC-ε, and PKC-ζ, play pivotal roles in regulating cellular processes such as growth, apoptosis, and neoplastic transformation. Montalvo et al. demonstrated that in PC3 cells, flutamide enhances the expression of these four PKC isoenzymes under nearly all experimental conditions tested. Notably, this effect of flutamide in PC3 cells is independent of the androgen receptor (AR), as no AR expression was detected in these cells (9). Despite the likely involvement of alternative signaling pathways in the action of flutamide on PC3 cells, it was observed that much higher doses of flutamide are required to achieve effects comparable to those observed in LNCaP cells. This suggests that flutamide may not be the optimal treatment choice for targeting PC3 cells. Therefore, further investigation into alternative therapeutic strategies or more effective antiandrogens may be warranted for this particular cell line.

However, several concerns exist regarding the use of flutamide. One major issue is the development of acquired resistance to flutamide, which is a common occurrence in patients with castration-resistant prostate cancer (CRPC) (10, 11). Although the exact mechanisms underlying this resistance are not fully understood, studies have indicated that point mutations in the androgen receptor (AR), such as T887A and W741C, may play a crucial role in the resistance to flutamide (12). Additionally, LNCaP cells that overexpress a mutated form of AR have been found to adapt the AR signaling pathway, enabling these cells to continue growing and surviving even under conditions of hormone therapy. This adaptation underscores the complexity of treatment resistance in prostate cancer (13, 14). Another significant drawback of flutamide is its range of side effects, which can severely diminish the quality of life for patients. These include liver toxicity, gastrointestinal disturbances (such as nausea, vomiting, and diarrhea), hot flashes, and gynecomastia. These adverse effects can be distressing for patients and may lead to treatment discontinuation in some cases (15).

To address the previously mentioned issues and challenges, we propose combining SeNPs with flutamide. SeNPs have demonstrated anticancer properties in prior studies (16, 17). Specifically, research by Sonkusre et al. has shown that SeNPs exhibit significant toxicity against PC-3 cancer cells, suggesting that their combination with flutamide could enhance treatment outcomes in this cell line (18). Additionally, Kong et al. have demonstrated that SeNPs can inhibit the growth of LNCaP cancer cells (17). Importantly, SeNPs are noted for their excellent biocompatibility and low toxicity, alongside their rapid absorption and substantial therapeutic effects. (19-23). These attributes make them promising candidates for combination therapy with flutamide, potentially leading to improved efficacy in targeting prostate cancer cells.

The precise molecular mechanisms through which SeNPs exert tumor-suppressive effects remain incompletely characterized. It is generally believed that SeNPs can induce apoptosis in tumor cells by promoting cellular uptake and mitigating reactive oxygen species (ROS) (24). Huang et al. has demonstrated that SeNPs can trigger autophagy in cancer cells, contributing to their anticancer effects (25). Additionally, a study by Sonkusre reported that SeNPs lead to the upregulation of TNF, which can induce necrosis in cancer cells (18). Vekariya et al. showed that SeNPs inhibit cell proliferation and hinder the synthesis of DNA, RNA, and proteins, indicating that they might alter the expression of various functional molecules, including non-coding RNAs (26). Furthermore, SeNPs have been found to induce cell cycle arrest and enhance apoptosis in prostate cancer cells. Studies usi

---

## [Decision Letter · Decision Letter 1]

17 Jan 2025

Reducing the Effective Dosage of Flutamide on Prostate Cancer Cell Lines through Combination with Selenium Nanoparticles: An In-Vitro Study

PONE-D-24-41345R1

Dear Dr. aghamir,

We’re pleased to inform you that your manuscript has been judged scientifically suitable for publication and will be formally accepted for publication once it meets all outstanding technical requirements.

Kind regards,

Ahmed E. Abdel Moneim

Academic Editor

PLOS ONE

Additional Editor Comments (optional):

Reviewers' comments:

Reviewer's Responses to Questions

**Comments to the Author**

1. If the authors have adequately addressed your comments raised in a previous round of review and you feel that this manuscript is now acceptable for publication, you may indicate that here to bypass the “Comments to the Author” section, enter your conflict of interest statement in the “Confidential to Editor” section, and submit your "Accept" recommendation.

Reviewer #1: All comments have been addressed

Reviewer #2: All comments have been addressed

2. Is the manuscript technically sound, and do the data support the conclusions?

Reviewer #1: Partly

Reviewer #2: Yes

3. Has the statistical analysis been performed appropriately and rigorously? 

Reviewer #1: I Don't Know

Reviewer #2: Yes

4. Have the authors made all data underlying the findings in their manuscript fully available?

Reviewer #1: (No Response)

Reviewer #2: Yes

5. Is the manuscript presented in an intelligible fashion and written in standard English?

Reviewer #1: Yes

Reviewer #2: Yes

6. Review Comments to the Author

Reviewer #1: (No Response)

Reviewer #2: (No Response)

7. PLOS authors have the option to publish the peer review history of their article (what does this mean? ). If published, this will include your full peer review and any attached files.

**Do you want your identity to be public for this peer review?** For information about this choice, including consent withdrawal, please see our Privacy Policy .

Reviewer #1: No

Reviewer #2: **Yes: ** Shimaa Elsayed Rashad Elsayed

---

## [Editor Report · Acceptance letter]

PONE-D-24-41345R1

PLOS ONE

Dear Dr. Aghamir,

I'm pleased to inform you that your manuscript has been deemed suitable for publication in PLOS ONE. Congratulations! Your manuscript is now being handed over to our production team.

Kind regards,

on behalf of

Dr. Ahmed E. Abdel Moneim

Academic Editor

PLOS ONE